# *Alnus glutinosa* Threatened by Alder *Phytophthora*: A Histological Study of Roots

**DOI:** 10.3390/pathogens10080977

**Published:** 2021-08-03

**Authors:** Corina Nave, Juliette Schwan, Sabine Werres, Janett Riebesehl

**Affiliations:** 1Julius Kühn Institute (JKI)–Federal Research Centre for Cultivated Plants, Institute for Plant Protection in Horticulture and Forests, Messeweg 11/12, 38104 Braunschweig, Germany; corina.nave@julius-kuehn.de (C.N.); swerres@t-online.de (S.W.); 2Julius Kühn Institute (JKI)–Federal Research Centre for Cultivated Plants, Institute for National and International Plant Health, Messeweg 11/12, 38104 Braunschweig, Germany; juliette.schwan@julius-kuehn.de

**Keywords:** clade 7, plant pathology, thin sections

## Abstract

Alder dieback remains a major problem in European alder stands and its spread continues to threaten their existence. The causal agent of this disease is the so-called alder *Phytophthora* species complex, which includes the hybrid *Phytophthora* ×*alni* and its parental species *P. uniformis* and *P. ×multiformis*. Little is known about the survival of these *Phytophthora* species in alder. The aim of our investigations was to find out whether, and if so where, the pathogen survives. The subject of these studies was alder roots. Therefore, artificial infection studies and histological studies with *P*. ×*alni* and *P. uniformis* were carried out on seedlings of black alder (*Alnus glutinosa*). These histological studies revealed oogonia and oospores of *P*. ×*alni* and *P. uniformis* in different parts of the root tissue.

## 1. Introduction

Common alder, also known as black alder (*Alnus glutinosa* ([L.] GAERTN.), is a very common tree species in Europe. Its occurrence ranges from the Mediterranean in the south and Ireland in the west to mid-Scandinavia in the north and western Siberia in the east. It is adapted to moderate to cold climates and prefers to grow in stands tied to water [1]. In this context, it plays a key role in the bank stabilization of water bodies. It also benefits aquatic ecosystems in numerous ways, including its ability to fixate nitrogen and its importance to various species as a source of food and shelter [2]. Black alders are almost unique in their role and cannot be easily replaced by any other tree species [3,4,5,6,7].

Since the early 1990s, dieback in the genus Alder has been observed throughout Europe. The first record of this issue was in the United Kingdom in 1993, followed by other European countries before it was detected in Germany in 1998—e.g., [8,9,10,11]. The disease primarily occurs near flowing waters. Affected alders show dead bark areas and mucilage flow at the base of the trunk due to cambium necrosis, as well as severe crown thinning before death. In this process, the alders often die within about three years of infection [12].

The disease is caused by the *Phytophthora alni* species complex, which consists of *P*. ×*alni* (Brasier and S. A. Kirk) Husson, Ioos and Marçais, nothosp. nov.; *P. uniformis* (Brasier and S. A. Kirk) Husson, Ioos and Aguayo, comb. nov.; and *P*. ×*multiformis* (Brasier and S. A. Kirk) Husson, Ioos and Frey, nothosp. nov. [13,14]. *P.* ×*alni* is the result of a hybridization event between *P. uniformis* and *P*. ×*multiformis* [14,15,16] and is the most aggressive of the three oomycetes [17]. The alder *Phytophthora* is extensively distributed in Central Europe [5]. These alders belong to clade 7 in DNA-based phylogenies, according to the key of Waterhouse [18], and are root pathogens, like most representatives in this group [19].

These are host-specific pathogens on alder trees, that, like all *Phytophthora* species, are perfectly adapted to water. Their zoospores are considered the main dispersal organs, and they are able to actively swim to host plants with the help of their two flagella [20]. They are also able to form oospores [12]. With these resting spores, the pathogens can survive for years even without a host plant—e.g., in the soil [21].

There has been no previous evidence of oospores in the roots of alder trees, let alone the presence of the pathogen in this tissue in general. Therefore, inoculation experiments with *P*. ×*alni* and *P. uniformis* were carried out on black alder seedlings and the roots were subsequently examined histologically. The purpose of this study was to investigate whether the pathogens are present in the roots after inoculation, which tissues they colonize, and whether they are able to form resting spores. The results show that both species are capable of penetrating the roots; spreading in the tissue; and forming oogonia, oospores, and sporangia.

## 2. Results

At the end of the inoculation studies, several root samples were taken per category (0 to 4) and examined histologically. The results for the health status and level of damage of the plants are not presented here.

### 2.1. Non-Inoculated Alders (Negative Controls)

The roots of the non-inoculated plants showed healthy tissue with normally developed cells. No foreign structures, such as mycelium or spores of oomycetous or fungal origin, could be found. The cells were stained middle to light blue, except for the epidermis. These cells showed a brown to yellow coloration (Figure 1).

### 2.2. Alders Inoculated with Phytophthora ×alni

The roots from the plants of category 0 (Figure 2A–C)—i.e., plants without symptoms—were found to have some cells of the vascular cylinder (vc) stained blue to green or brown (Figure 2A). In the peripheral area of the cortex, an oogonium with an oospore inside was detected intracellularly. Mycelium was also seen adjacently (Figure 2A). The cross-section of another sample showed an oogonium intercellularly. All cells showed a middle to dark blue coloration (Figure 2B). Figure 2C shows a part of the roots where the cutting process damaged the cells. Nevertheless, oogonia (stained blue) and hyphal swelling (stained yellow) are visible within the cells (Figure 2C).

All the root cuts classified into category 1 showed structures of *Phytophthora* (Figure 2D–G). In the cortex of one sample, a sporangium prior to differentiation is lying outside the cells, possibly due to the tearing of the specimen during cutting (Figure 2D). In the same sample, mycelium was detected within the cells, marked with a red arrow. The root cells in Figure 2F,G appear to be healthy; however, intracellular oogonia and mycelium could be observed. In contrast to this, there were also samples in which the cell structure already showed severe damage, despite the fact that the plants showed hardly any symptoms, as shown in a longitudinal section with several intercellular bullate oogonia with oospores (Figure 2G).

In the evaluation of the infection trials, category 2—i.e., plants with the first wilting symptoms—did not occur. Accordingly, this category is not described.

Figure 2H–K are examples of thin root sections of category 3 plants. The three longitudinal sections clearly show an undamaged cell complex. All cuts show intracellular bullate oogonia with oospores in the cortex (Figure 2H,J,K), as well as in the vascular cylinder (vc) (Figure 2I).

Category 4 summarizes all the plants that have died as a result of inoculation. Looking at the sections from the roots of these plants, *P.* ×*alni* did not cause severe damage to the cell complex. Again, structures such as oogonia were detected close to the vascular cylinder (vc) (Figure 2L), intracellularly in the outer cortex (Figure 2M), or between the cells at the border of the vascular cylinder (vc) (Figure 2N). Several oospore formations were observed, often intracellular, and in some cases, mycelium was observed close to the oogonium (Figure 2O, red arrow).

### 2.3. Alders Inoculated with Phytophthora uniformis

The cells of root sections from plants inoculated with *P. uniformis* were stained differently. Cells of the cortex showed a light blue and sometimes greenish color. The vascular cylinder was dark blue or brownish in color, as was the epidermis if it was present in the sections (Figure 3).

In the root cuts of category 0, which contained plants without any symptoms, an oogonium was detected in the longitudinal cut. It was located intercellularly very close to the cells of the vascular cylinder and had a blue to green color (Figure 3A). No oomycete structures were found in the cross-section. The greenish to blue and brown coloration in the outer area of the cortex differed from the other cells of the cortex, which were stained blue. The cells were undamaged (Figure 3B).

The longitudinal root sections from plants of category 1 showed several oomycetous structures in the periphery of the cortex. They were colorless with blue outlines and were located both intracellularly and intercellularly. The cell association seemed intact and showed hardly any damaged cells. Some of the oogonia had already begun to differentiate (Figure 3C–E).

In the evaluation of the infection trials, category 2—i.e., plants with the first wilting symptoms—did not occur. Accordingly, this category is not described.

Figure 3F–J show cross and longitudinal sections of roots from category 3 plants. The cell association was largely undamaged. Structures of *Phytophthora* were not detectable in the vascular cylinder or in the epidermis. In the cross-sections (Figure 3G,I), all oogonia were observed intracellularly in the cortex, partly with mycelium (Figure 3I). The cuts in a longitudinal direction showed different structures of *P. uniformis* (Figure 3F,H,J). Figure 3F shows a single oogonium, which was located intracellularly, starting to penetrate a neighbor cell. Figure 3H shows intracellular oogonia with mycelium in the cortex, where one penetrates the neighboring cell (red arrow). Zoospores could be detected in a section with damaged cells, most of them intracellularly.

Most roots of category 4 plants were in good condition (Figure 3K,L). A lot of intracellular mycelium (Figure 3K) and many oogonia, some of them with oospores (Figure 3L), were found. The latter were located inter- and intracellularly in heavily damaged cells of the cortex.

## 3. Discussion

In this study, an artificial inoculation method was used to infect alder seedlings with alder *Phytophthora*, followed by the histological examination of the roots for possible tissue colonization. Histology results clearly show that the pathogens, *P.* ×*alni* and *P. uniformis*, successfully colonized the root tissue. Structures such as mycelium, sporangia, zoospores, oogonia, and even oospores could be identified. These structures were most frequently found intra- and intercellularly in the cortex, but in some cases also in the vascular cylinder. These results are unique so far. Other studies have already shown that when the pathogen reaches the alder with the help of its zoospores, it is able to germinate and form mycelium that grows into the host tissue. From here, the pathogen can continue to grow and reproduce vegetatively [20,22]. Infection occurs, for example, via adventitious roots or lenticels [23]. *P.* ×*alni* and *P. uniformis* belong to the subclade 7a, where all associated species are mostly pathogenic on the roots of their host [19,24].

Furthermore, both *Phytophthora* species mainly grew intracellularly. This has also been shown by many other studies with *Phytophthora* species in roots [25,26,27,28]. In this study, we showed that in most cases even the formation of the oogonia took place in the intracellular space. It also seems that the cortex is favored by the pathogen over the epidermis and vascular cylinder. However, the latter is also better protected from penetration by reinforced cell walls.

The results of pathogenicity tests conducted by Brasier and Kirk [17] are not reflected in the present study. In contrast to their observations, in this histological study, both isolates showed a high potential to infect the cells and spread within them. Interestingly, both pathogens could be detected in samples from different plant health categories, from healthy, symptom-free plants (category 0) to dead plant tissue (category 4). This indicates that *P*. ×*alni* and *P. uniformis* can establish in living as well as in damaged cells. This is reflected by the fact that many *Phytophthora* species have a hemibiotrophic lifestyle. They start infections biotrophically in living plant tissues but transform to a parasitic existence later on. They continue to spread and potentially form resting propagules or sporulating structures in damaged or dead cells [21,29,30,31]. These observations correlate with infected alders in their natural habitat. According to Werres et al. [3] and Jung and Blaschke [23], many years can pass before the first symptoms, such as bleeding cankers, become visible due to the infection. The reasons for this can be diverse, including high host resistance and unfavorable environmental or living conditions. Another reason could be a good strategy on the part of the pathogen. Oomycetes can survive in the soil or plant remains for a long time [30]. This strategy includes, as we observed, the formation of resting structures, necessary to ensure survival over time. Oospores are not considered important in spreading under natural conditions [11,32,33]. Detailed studies showed different viability rates and no germination of oospores from natural alder *Phytophthora* variants [33]. However, these studies with in vitro cultures also showed high rates of degeneration of the cultures studied. In addition, the studies on oospore viability and germination rate were exclusively conducted in vitro. In our studies, the sole aim was to show whether oospores can be formed in living tissue. Thus, it cannot be excluded at this point to what extent the spores are survivable and germinable. An important aim of new studies with oospores in living plant tissue should be to show whether these naturally produced oospores are viable. As shown for *P. cinnamomi* and *P. palmivora* infecting Medicago roots, different structures, such as mycelium, chlamydospores, and zoosporangia, are formed within the root tissue after penetrating it via haustoria [28]. Some *Phytophthora* species are able to produce haustoria; other oomycetes probably use alternative mechanisms to penetrate the cells [34]. No haustoria were observed in this study.

The *Phytophthora alni* species complex is primarily responsible for the alder decline, additionally facilitated by human activity. One factor identified as the main cause for the introduction or spread of the pathogens is infected nurseries—e.g., [35,36]. Jung et al. [21] suggest that they are the major source of *Phytophthora* spread into the environment, artificially accelerating the spread by planting infested nursery stock. Other studies blame the intensified international plant trade as a major cause of invasive forest disease introductions [37,38,39], which results in further risk. Random encounters of closely related allopatric *Phytophthora* species, which have not established reproductive barriers due to geographic separation, can easily allow them to hybridize, as has happened in the case of *P*. ×*alni* [13,14,21].

A few years ago, this hybrid species was included in the list of the 20 most important oomycetes in molecular plant pathology. This was due to the scientific and economic importance of *P.* ×*alni* [40]. This highlights once again the negative impact of this pathogen on its environment and shows how urgently further studies are needed.

Despite studies showing that the three alder *Phytophthora* species are not pathogenic towards other tree species [17,41], it is known that other tree species play an important role in the survival of these pathogens. The alder *Phytophthora* species can survive in other tree species, such as sweet chestnut, walnut, and wild cherry [41,42,43]. It would be of great importance to understanding the spread and survival of the pathogen to show how it survives in these types of intermediate hosts, and why these trees are apparently not sensitive to the *Phytophthora* species.

Furthermore, it would be of interest to study other alder species, such as Italian alder (*Alnus cordata* (Loisel.) Duby), red alder (*Alnus rubra* Desf. ex Steud), green alder (*Alnus alnobetula* (Ehrh.) K.Koch), and grey alder (*Alnus incana* (L.) Medik.), in which *P*. ×*alni* and *P. uniformis* can also cause severe symptoms and are able to spread intracellularly [15,41,44,45].

This research points out that a key to understanding the processes that take place during infection is the histology of the affected tissue. This provides basic information about how the pathogen behaves in the tissue, how it can survive in the tissue, and which recognizable defense mechanisms might exist on the plants. This knowledge could help us to understand and prevent the spread of pathogens.

## 4. Material and Methods

### 4.1. Phytophthora Isolates and Inoculum Production

The *Phytophthora* strains (Table 1) were cultivated on carrot-piece agar at 20 °C in the dark according to Pogoda and Werres [46]. As soon as the entire petri dish was overgrown with mycelium (92 mm diameter), it was flooded to induce the production of sporangia and the release of zoospores. For flooding, 15 mL of an extract made of white peat and tap water was used. To obtain the extract, white peat (pH 4.5 ± 0.5) was carefully mixed with tap water in a volume ratio of 1:2 and left for 20 min at room temperature. The mixture was then filtered through a pleated filter (MN 605 ¼, diameter 270 mm). The flooded *Phytophthora* cultures were incubated for 3 days at 20 °C with 16 h of light. After this time, the liquid was exchanged with fresh peat extract. Following another day at 20 °C and 16 h of light, the cultures were incubated for 1 h at 4 °C and then 1 h at room temperature. The obtained zoospore suspension was diluted to 1 × 10^4^ zoospores/mL and used immediately for the infection studies. For the negative control, only the extract made out of white peat and tap water was used.

### 4.2. Alder Seedling Cultivation

The infection studies were conducted with *Alnus glutinosa* seedlings. Before sowing, the seeds were soaked in water overnight. Preliminary tests have shown that this increases the germination rate. The next day, the seeds were collected in a sieve and placed on filter paper, which absorbed excess moisture.

The sowing trays were filled with a mixture of clay substrate and sand (ratio 3:1) and the seeds were evenly distributed on the top and lightly covered with the substrate mixture. The trays were covered with a lid and placed in a climatic chamber set to 20 °C and 16 h of light.

At three weeks of age, seedlings were pricked into cultivation trays (QuickPot QPD 144/6R, Hermann Meyer KG, Rellingen, Germany). After another two months in the climatic chamber at 20 °C and 16 h of light, the infection trials with the seedlings began.

### 4.3. Infection Studies

One week prior to inoculation, the cultivation conditions in the climatic chamber were changed to 20 °C during the day and 12 °C at night with a sinusoidal temperature progression and 12 h of light. These settings were maintained for the entire infection trial and imitated the climatic conditions of late summer because host susceptibility is highest from June to September [47].

For inoculation, large containers were each filled with 8 L of tap water, 250 mL of zoospore suspension, and 8 of the Petri dishes with Phytophthora cultures used for the preparation of the zoospore suspension. For the negative control, tap water, white peat extract, and eight Petri dishes with carrot-piece agar were used. The cultivation trays with the plants were placed in these containers with the root balls completely submerged. The high water level in the container imitated the flooding of the root area as it occurs on banks and enabled the zoospores to penetrate the roots. After three days, the containers with the inoculum were exchanged for containers with pure tap water and watered regularly to maintain a high water level.

The final assessment took place three weeks after inoculation and included the classification of the seedlings into five categories, depending on their health status and level of damage (Table 2).

### 4.4. Histology and Microscopy

For the histological studies of the alder roots, the methods of Pogoda and Werres [46] for sample processing, pre-infiltration, infiltration, polymerization, sectioning, and staining were adapted for root tissue.

The root samples of infected seedlings from the different categories (0–4), as well as non-inoculated roots, were placed in a formalin-acetic-acid-alcohol solution for fixation. One hundred milliliters of this solution was composed of 23.9 mL formaldehyde (37%), 32.6 mL ethanol (96%), 10.9 mL acetic acid (100%), and 32.6 mL distilled water. The samples were held under vacuum two to three times for 5 min at a maximum of 100 mbar and then stored overnight at 4 °C. This was followed by dehydration in an ascending series with isopropanol (Table 3).

For embedding, the Technovit^®^ 7100 system (Heraeus Kulzer GmbH, Wehrheim, Germany), a plastic embedding system based on HEMA (2-hydroxyethyl methacrylate), was used.

The solutions used for pre-infiltration and infiltration were prepared according to the manufacturer´s instructions. For pre-infiltration, the samples were incubated twice for 5 min under vacuum in the pre-infiltration solution and then stored for at least 2 h in the same solution at room temperature. After changing the solution for the infiltration step, incubation was again performed twice for 5 min under vacuum. The samples remained in the infiltration solution overnight at 4 °C.

The solution for polymerization was prepared according to the manufacturer’s instructions and poured into a mold (Histoform S by Heraeus Kulzer GmbH, Wehrheim, Germany). The infiltrated samples were gently placed in the mold and adjusted. The polymerization was carried out at room temperature for 1 h, followed by 1 h at 37 °C. After complete curing, the specimens were mounted on plastic blocks (Histoblocs) using Technovit^®^ 3040 (Heraeus Kulzer GmbH, Wehrheim, Germany).

Thin sections with a thickness of 10 µm were cut with a rotary microtome (HM 355, MICROM international GmbH, Dreieich, Germany). These thin sections were stained for 3 min with toluidine blue (0.03 g toluidine blue in 100 mL SOERENSEN’s Buffer (content: Na2HPO4 2 H2O: 10.68 g/L and KH2PO4: 5.44 g/L, pH 7.0; Morphisto GmbH, Offenbach am Main, Germany)) and covered with Entellan^®^ (Merck KGaA, Darmstadt, Germany) for long-term storage.

For analyzing the samples, a Zeiss microscope Axio Imager M2 and the software Zen 2 (Carl Zeiss AG, Oberkochen, Germany) were used.

## 5. Conclusions

The histological studies of the alder roots showed that both *Phytophthora* species can develop gametangia in the roots. Oogonia and oospores could be observed in susceptible and less susceptible root tissue.

## Figures and Tables

**Figure 1 pathogens-10-00977-f001:**
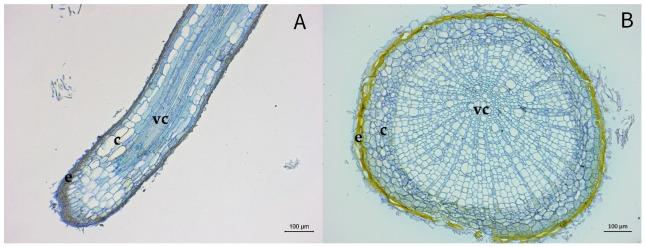
Thin sections of non-inoculated roots from *Alnus glutinosa*. Epidermis (e), cortex (c), and vascular cylinder (vc). (**A**), longitudinal section. (**B**), cross-section.

**Figure 2 pathogens-10-00977-f002:**
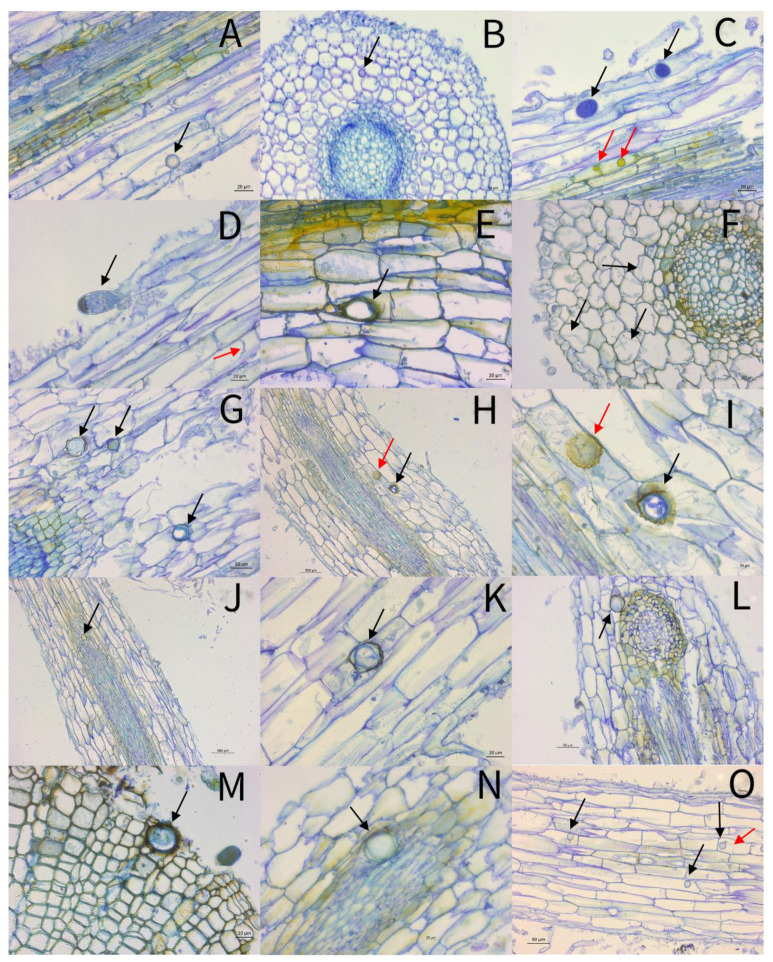
Thin sections of roots from *Alnus glutinosa* inoculated with *Phytophthora* ×*alni* three weeks after inoculation. (**A**) plant health status category (phsc) 0, longitudinal section, intracellular oogonium with oospore in the cortex (black arrow). (**B**) Phsc 0, cross-section, intercellular oogonium (black arrow). (**C**) Phsc 0, longitudinal section, several intracellular oogonia (black arrows), and hyphal swellings (red arrows). (**D**) Phsc 1, longitudinal section, sporangium close to damaged cortex cells (black arrow), and mycelium intracellular (red arrow). (**E**) Phsc 1, longitudinal section, intracellular oogonium with antheridium (black arrow). (**F**) Phsc 1, cross-section, intracellular mycelium in the cortex (black arrows). (**G**) Phsc 1, longitudinal section, intercellular bullate oogonia with oospores (black arrows). (**H**) Phsc 3, longitudinal section, intracellular oogonia with oospore in the cortex (red and black arrow). (**I**) enlarged image section of (**H**): intracellular bullate oogonium with oospore (black arrow) and an intracellular not differentiated bullate oogonium (red arrow), both close to a vascular cylinder. (**J**) Phsc 3, longitudinal section, oogonium with oospore in a vascular cylinder (black arrow). (**K**) Phsc 3, longitudinal section, intracellular bullate oogonia with oospore (black arrow). (**L**) Phsc 4, longitudinal section, oogonium with antheridium in growth tissue (black arrow). (**M**) Phsc 4, cross-section, oogonium with antheridium in the cortex (black arrow). (**N**) Phsc 4, longitudinal section, oogonium with oospore in a vascular cylinder (black arrow). (**O**) Phsc 4, longitudinal section, several oogonia with oospores intracellular in the phloem (black arrows), and oogonium with developing mycelium (red arrow).

**Figure 3 pathogens-10-00977-f003:**
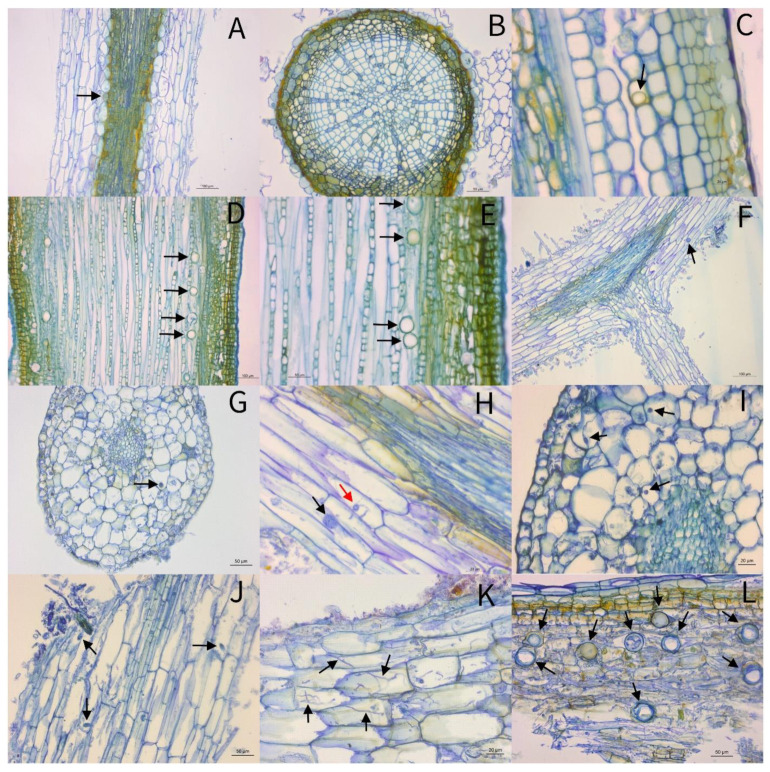
Thin sections of roots from *Alnus glutinosa* inoculated with *Phytophthora uniformis* three weeks after inoculation. (**A**) plant health status category (phsc) 0, longitudinal section, oogonium stained blue to green, intercellular very close to the vascular cylinder (black arrow). (**B**) Phsc 0, cross-section, no *Phytophthora* structures found. (**C**) Phsc 1, longitudinal section, oogonium intracellular in the cortex (black arrow). (**D**) Phsc 1, longitudinal section, several intercellular oogonia in the cortex (black arrows). (**E**) Phsc 1, longitudinal section, several intercellular oogonia in the cortex (black arrows). (**F**) Phsc 3, longitudinal section, intracellular undifferentiated oogonium in the cortex (black arrow). (**G**) Phsc 3, cross-section, intracellular oogonium (black arrow). (**H**) Phsc 3, longitudinal section, intracellular germinating oogonia in the cortex (black arrows), one of them penetrating the neighbor cell (red arrow). (**I**) Phsc 3, cross-section, intracellular oogonia, and mycelium in the cortex (black arrows). (**J**) Phsc 3, longitudinal section, inter- and intracellular zoospores in the cortex (black arrows). (**K**) Phsc 4, longitudinal section, intracellular mycelium in the cortex (black arrows). (**L**) Phsc 4, longitudinal section, several inter- and intracellular oogonia, some of them with oospores in the heavily damaged cortex (black arrows).

**Table 1 pathogens-10-00977-t001:** *Phytophthora* Species.

*Phytophthora* Species	Isolate Number	Isolated From	Country of Origin	Year of Isolation
*P. ×alni*	JKI-026-15-8-01-2-0	soil	Germany	2005
*P. uniformis*	JKI-005-16-8	soil	The Netherlands	1998
*P. uniformis*	BBA 7/03	soil	Germany	2002

**Table 2 pathogens-10-00977-t002:** Plant health status in categories.

Plant Health Status	Category
Plant with no symptoms	0
Leaves begin to dry out	1
First wilting symptoms	2
Dried up leaves + wilting symptoms, sometimes necrotic shoots	3
Death of the plant	4

**Table 3 pathogens-10-00977-t003:** Dehydration steps with isopropanol.

Isopropanol	Vacuum	Waiting Time
50%	2 × 5 min	20 min
50%	2 × 5 min	20 min
70%	2 × 5 min	20 min
90%	2 × 5 min	20 min
96%	2 × 5 min	20 min
99%	2 × 5 min	20 min

## Data Availability

Data supporting the conclusions of this article are included within the article.

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
