# Peer review of "Alnus glutinosa Threatened by Alder Phytophthora: A Histological Study of Roots"

_pathogens, 2021, doi:10.3390/pathogens10080977_

Round 1

Reviewer 1 Report

The manuscript “ Alnus glutinosa threatened by alder Phytophthora – a histologi-2 cal study of roots” by Nave & al report an inoculation study of A. glutinosa seedling with P. xalni and P. uniformis. Histological observation were done on inoculated seedling roots and it was shown that oospores are formed by both pathogens in alder root tissus.

The topic is of interest as P. xalni is a major pathogen in riparian ecosystems and it ability to survive with oospores has been debated. Showing that some are produced in roots of infected alder is a step further worth publishing. However I feel that the manuscript push in conclusions that can’t really be reached.

A strong point is that oospore formed by alder Phytophthora sensus lato (standard variant, probably P. xalni) in vitro were shown to be not viable (Delcan & Brasier, 2001, 10.1046/j.1439-0329.2001.00223.x). This goes with the fact that P. xalni is a triploid hybrid (Husson & al, 2015) and thus potentially might not be able to produced viable sexual spores. Moreover, this pathogen has been shown to persist little in soil in natural situations (Jung, T.; Blaschke, 2004, Elegbede & al, 2006, 10.1094/PHYTO-05-10-0140). Some of those papers are not even cited, in particular Delcan & Brasier (2001), which is very poor.

It remains very debatable that viability of oospores produced in vitro well represent what happens in natural conditions in soil/root. Thus showing that oospore are indeed produced in roots is very valuable. However, I do not feel it was not shown that the oospore produced in alder roots were viable. If the authors feel they did show that, they have to explain why. Ignoring the debate is not acceptable. This should be a big point in the discussion.

Minor remarks.

L44-45. The sentence seems awkward

L243 tap and not tab?

L265 & 268. I missed something. I do not understand what ‘clones’ refers to.

L97 « Severel». Several ?

L140 a red arrow

L194-197. These 2 sentences are not clear. I do not understand what is said. Ok, it takes time from infection to first symptoms (canker, crown thinning). But, this may be interpreted in many ways: unfavorable environment, poor aggressivety / high host resistance for example. It do not necessarily means a “well-established strategy”. Do you mean that it show that P. xalni/uniformis are able to survive during this time and thus should have the means for this?

Then, maybe symptom development takes some years because survival is poor which results in low inoculum build-up and symptom development need specific environmental conditions leading to very brutal inoculum build-up.

It is difficult to speculate from that fact …  

L200. The citation of 10.1046/j.1365-3059.2001.00553.x for this assertion is spurious. In this work, Brasier & Kirk test the aggressivity of isolates from the P. alni complex by measuring the lesion length induced in 1-2 month on alder log. I do not see anything on oospore, and even more on oospore germination … The adequate citation is Delcan & Brasier (2001) that point out that oospores produced are not viable !

Reviewer 2 Report

the manuscript title "Alnus glutinosa threatened by alder Phytophthora – a histological study of roots" contains interesting information about the plant-pathogen interaction and analysis. The manuscript represents the detailed protocol to do histological studies of the alder roots and Phytophthora species that can develop gametangia in the roots. Moreover, the author discussed that fungal oogonia and oospores could be observed in susceptible and less susceptible root tissue. 

Strength  

1). Author discussed histological studies with P. ×alni 16 and P. uniformis and seedlings of black alder (Alnus glutinosa).

Weakness

1)      The used methods are very basic and the article should be published as a short report, not like an article.

2)      To be an article, the author needs to add validation part by using anatomical studies by using some dye under the fluorescent light microscope.

3)      An SEM analysis can also help to validate the plant-pathogen interaction.

Summary: The study is very basic, need to add more evidence about the result and methodology.

Round 2

Reviewer 1 Report

The authors strongly improved the manuscript and I feel it can be accepted in it present form

A minor remark:

L207: "it cannot be excluded". Is excluded the correct word ? maybe more inferred ?

Reviewer 2 Report

The author justified all of the comments. the manuscript is ready for publication.